# The perceptions of different professionals on school absenteeism and the role of school health care: A focus group study conducted in Finland

Katja Melander[1]*, Tiina Kortteisto[2], Elina Hermanson[3], Riittakerttu Kaltiala[4], Katariina Mäki-Kokkila[5], Minna Kaila[6], Silja Kosola[7]

1 University of Helsinki, Helsinki, Finland, 2 Tampere University Hospital, Tampere, Finland, 3 Pikkujätti Medical Center for Children and Youth, Helsinki, Finland, 4 University of Tampere, Tampere, Finland, 5 City of Kirkkonummi, Kirkkonummi, Finland, 6 University of Helsinki /Public Health Medicine, Helsinki, Finland, 7 Pediatric Research Center, New Children's Hospital, Helsinki University Hospital and University of Helsinki, Finland

* katja.melander@helsinki.fi

**Data Availability Statement:** All relevant data are within the paper and its Supporting Information files.

## Abstract

### Purpose of the study

School absenteeism and school dropout jeopardize the future health and wellbeing of students. Reports on the participation of school health care in absenteeism reduction are infrequent, although physical and mental health problems are the most common causes of school absenteeism. Our aim was to explore what reasons different professionals working in schools recognize for absenteeism and which factors either promote or inhibit the inclusion of school health care in absenteeism reduction.

### Materials and methods

Data for this qualitative study was gathered from ten focus groups conducted in two municipalities in southern Finland. The groups included (vice) principals, special education/resource/subject teachers, guidance counselors, school social workers, school psychologists, school nurses, school doctors, and social workers working in child protective services. Data analysis was predominantly inductive but the categorization of our results was based on existing literature.

### Results

Study participants identified student-, family-, and school-related reasons for absenteeism but societal reasons went unmentioned. A number of reasons promoting the inclusion of school health care in absenteeism reduction arose, such as expertise in health-related issues and the confidentiality associated with health care. Inclusion of school health care was hindered by differences in work culture and differing perceptions regarding the aims of school health care.

**Funding:** KM (the main researcher) has been funded by grants. The funders haven't had any role in any stage of the study or the preparation of the manuscript. The funders are: - Niilo Helander foundation https://www.niilohelander.net/ - The Finnish Cultural Foundation https://skr.fi/en.

**Competing interests:** The authors have declared that no competing interests exist.

## Conclusion

Professionals working in schools were knowledgeable about the different causes of school absenteeism. Clarifying both the aims of school health care and the work culture of different professionals could facilitate the inclusion of school health care in absenteeism reduction.

## Introduction

School absenteeism and potential subsequent school dropout have been called a public health threat [1–3] because they may jeopardize student health and development in multiple ways [3, 4]. Absenteeism has a negative effect on school performance [5] which may lead to a lower educational level and inferior health in adulthood [6]. Absenteeism associates with risky behavior such as substance use, risk-taking in traffic, and ill-considered sexual behavior which can negatively affect the health of a student [7]. Absenteeism is also associated with difficulties in social relationships with peers [8].

Literature discerns two types of absenteeism: excused and unexcused. Excused absences occur when the student has permission from a parent/guardian to be absent from school, for example due to an illness or a family event. When the student is absent without permission the absence is unexcused and the student is truant. For more than a decade, truancy has been decreasing in developed countries in contrast to increasing excused absences [7, 9, 10]. Absenteeism can also be categorized based on an ecological model into student, family, school, and society level reasons for absenteeism [11–14].

Teachers are often required to intervene in absenteeism before school health and welfare professionals [15]. Thus, they need to be conscious of the possible reasons for absenteeism. Previous studies have contradictory findings on how adept teachers actually are in recognizing the reasons behind absenteeism. In a US study conducted in 2011, school personnel had difficulties in discerning the different causes of absenteeism [16], whereas in a Swedish study, teachers recognized that school absenteeism has multiple origins, with student- and family-related reasons the most commonly named [17]. This study aims to increase the knowledge on the reasons that professionals working in schools recognize for absenteeism. We focus on problematic absenteeism as defined by Kearney [18]: the student is absent at least half of their lessons during a two-week period and/or has such trouble attending school regularly during a two-week period that either the life of the student and/or their family is severely influenced.

A multinational comparison discovered that school health care (SHC) is most effective when school-based and with a multi-professional staff dedicated solely to SHC work [19]. Internationally, one goal of SHC is to tackle health issues inhibiting regular school attendance of students [20]. Multiple health-related reasons for absenteeism [3] include mental health problems and undiagnosed or poorly managed chronic illnesses such as asthma [1, 21–23]. Parents/guardians have also stated that health reasons are the most common motive they allow the student to be absent [24]. Intuitively, SHC could offer valuable support to both students and teachers in reducing absenteeism. However, in Finland SHC may be overlooked in absenteeism reduction since municipality-level guidelines on absenteeism reduction variably mention SHC [25].

Health care professionals may in fact be "in a key position"[1] in absenteeism reduction and school reintegration. This statement is supported by studies where SHC measures have been able to reduce absenteeism [6, 26–28]. The aim of this study was to explore factors that either promote or inhibit the inclusion of SHC in absenteeism reduction. This study focuses

on students aged from 13 to 15 years, because this is the time when absenteeism often increases [29, 30], and absenteeism during this time period predicts the future academic success of the student [31–33].

## Materials and methods

We organized focus groups comprised of educational, school healthcare, and child protection professionals to explore their views on three questions:

1. what reasons do different professionals recognize for school absenteeism in 13-to-15-year-old adolescents,

2. why should SHC be included in absenteeism reduction, and

3. what inhibits the inclusion of SHC in absenteeism reduction in this age group?

### Study environment

**The Finnish school system.**   Finland has consistently performed well in Programme for International Student Assessment (PISA) surveys since the first results in 2001 [34], raising international attention to its educational system [35, 36]. Finnish municipalities are mandated by law to organize basic education [37, 38]. Thus, most Finnish schools are public; less than two percent of students attend state or private schools. The nine-year basic education begins the year a child turns seven, and it is succeeded by three years of secondary education. From the beginning of August 2021, compulsory education ends at age 18 or at completion of secondary education; previously compulsory education ended after basic education or at age 17.

Compulsory education is government-funded, as are school meals [37]. Schools are legally bound to monitor student absences and to inform parents/guardians of any unexcused absences, while parents/guardians are responsible for ensuring that their child completes compulsory education [38]. Most schools in Finland use web-based programs or applications to monitor absenteeism and to communicate with parents/guardians. Currently, each municipality can tailor its own guideline on when and how to intervene in absenteeism, resulting in varying intervention processes [25]. In Finland, the annual drop-out rates have been small: approximately 0.6% for basic education, 3.0% for academic secondary education and 9.4% for vocational secondary education [39]. Thus, the focus of this study was on problematic absences instead of drop-outs.

**School health and welfare services.**   School health and welfare services are offered to all students during compulsory education [40]. According to law, these services aim to promote general health and wellbeing, ensure a healthy learning environment, and provide early interventions when needed [41]. These services consist of a school psychologist, a school social worker and SHC, which includes a school nurse (a public health nurse by training) and a school doctor. Nationally recommended quotas are 800 students per psychologist and social worker, 600 students per school nurse, and 2100 students per school doctor [36]. These professionals meet regularly to discuss student wellbeing on a general level. In order to discuss the situation of a particular student, a tailored multi-professional team is assembled with the permission of the respective student and/or their parent/guardian [41].

SHC is part of the public primary health care system and cost-free for students. All students have regular contact with SHC as the school nurse is legally bound to examine each student annually, and the school doctor performs health checks in first, fifth, and eighth grade [42]. The Finnish Institute of Health and Welfare defines the aims of the check-ups. The aims include several screening measures (e.g. vision and scoliosis), preventive measures (e.g.

immunizations) and individual health promotion (e.g. discussions regarding sufficient sleep). The school doctor complements the screening and health promotion conducted by the nurse.

In comparison to SHC, school psychologists and school social workers focus primarily on school level work. They meet individual students only when needed, for example based on a referral from the school nurse. School psychologists concentrate on learning difficulties and mental health problems, whereas school social workers support the student in social interaction issues.

### Study design

**Study sites.** The study was conducted in two municipalities in Southern Finland. First, we organized two pilot focus groups in one school in Kirkkonummi (population 39,600, population of 7-15-year-olds 5300, 13% of total) [43, 44]. The pilot focus groups were organized to test the interview questionnaire, and to evaluate whether the focus groups should be hetero- or homogenous in terms of the professions of the participants. The school was chosen because one of the members of the research group (KM-K) worked there; however, she did not attend the focus group. Both pilot focus groups were organized at the school premises; the first was held instead of a routine meeting of the school welfare group, and the second was held on the same day after hours.

Helsinki, the capital of Finland (population 654,000, population of 7-15-year-olds 53700, 8% of total) was the main study site where eight focus groups were held [43]. Schools were invited to participate in the study based on the Positive Discrimination Index (PDI) [45]. This index takes into account the proportion of immigrants living in the school catchment area, parent/guardian educational level, and annual family income in the school district. The PDI is regularly updated and schools receive financial support based on their scoring. We e-mailed invitations to the five highest ranking and five lowest ranking schools based on the PDI updated in 2016. Seven schools expressed interest in the study, four of the lowest ranking and three of the highest ranking schools. One principal claimed that absenteeism was not an issue in their school and declined participation; two schools never responded to our invitation. Two schools originally interested in the study were unable to participate due to conflicting schedules. In total, five schools from Helsinki participated in the study: two low-ranking and three high-ranking schools.

Additionally, an open invitation was sent to school doctors via their chief, inviting them to a focus group specifically organized for them. Similarly, social workers from child protection services were invited with an open invitation sent via their chief. They had a choice between two dates for participation.

**Focus group procedure.** The ethics committee of the Hospital District of Helsinki and Uusimaa (HUS) decreed in December 2015 that the study could proceed since all study participants were voluntary adults who would provide written informed consent prior to participation.

Prior to scheduling the focus groups in Helsinki, one researcher (KM) met with either the principal or members of the school welfare group of each school. She also had a meeting with the chiefs of school doctors and social workers. The focus groups took place instead of the weekly meeting of the school welfare group. The focus groups for school doctors and social workers were held in a meeting room in central Helsinki after working hours.

Before each focus group, participants received written and oral information on the study and the research method. After receiving this information, the participants signed a written informed consent form. All groups were held in Finnish and all participants were anonymous in the recordings. No repeat interviews were organized. Two members of the research group (KM and TK) participated in all focus groups. TK acted as moderator, modifying the order of

questions and asking additional questions. KM acted as facilitator, responsible for the audio technique and field notes. KM and TK discussed data saturation after each focus group. The only people present during the focus groups were the participants, TK and KM. Participation was voluntary. The pilot focus groups took place in May 2016 and the proper focus groups from late 2016 to the end of 2017.

Focus groups were digitally recorded and then transcribed verbatim from January to February 2018. Of the ten recordings, eight were transcribed by a company (Tutkimustie Oy) and two by an independent entrepreneur. The participants did not comment or correct the transcripts, nor did they provide feedback on the findings.

**The discussion guide.**  A semi-structured topic guide was developed by KM with contribution from the research group, two members (TK and MK) of which had used this method previously. The order of the questions was interchangeable, and not all questions were posed during every focus group; however, the topics of the study questions were discussed during each group. No major changes were required to the topic guide after the pilot groups and the data gathered in these groups was also used in the analyses. The topic guide is included as S1 Appendix.

**Analysis process.**  All identified themes were derived from the recorded data. First, the data was read multiple times by three researchers (TK, SK and KM) independently. Every quotation answering one of the research questions was then isolated. Each quotation can be traced back to the original transcript with the use of the identification code referring to the group number whence the quotation came from (focus group, FG1-10). Isolation was first done individually and findings were then compared to ascertain that all relevant data had been gathered. Any discrepancies were rechecked to determine whether the problem was in the interpretation of the data or whether a relevant quotation had been overlooked.

After isolation, KM organized the quotations thematically, whereafter TK and SK verified the result. Data regarding the reasons for school absenteeism were first categorized according to existing literature and subcategories were then created. For example, the following quotation *"In some cases, something has happened at home, some family crisis which explains [the absenteeism]." (social worker, FG9)* was categorized to family-related reason for absenteeism, then to the "changes in the family"subcategory. Quotations regarding SHC were first organized according to the professional background: educational professionals (consisting of (vice) principals, special education/resource/subject teachers, guidance counselors, school social workers, and school psychologists), SHC professionals (including school nurses, and school doctors), and social workers. Secondly, the quotations were divided into reasons for including or excluding SHC, and finally into thematical subcategories. For example, the quotation: *"I personally consider the influence of school doctors and school nurses very minor in this whole field." (special education teacher, FG7)* was first categorized based on the profession of the participant and then as a general reason inhibiting the inclusion of SHC in absenteeism reduction. Reporting was based on the COREQ checklist [46].

## Results

The ten focus groups had 55 participants, one to eleven per group (Table 1). Most participants (75%) were female and the average duration of the focus groups was 63 minutes. The mood was relaxed and the participants were forthcoming with their views during the focus group sessions.

### Reasons for absenteeism

The reasons for absenteeism are presented in Table 2. For more details and quotations, please see S1 Data.

**Table 1. General information on the interviews.**

| | |
|---|---|
| **No. of groups** | 10 |
| **No. of participants** | 55 |
| **Females/males** | 41/13 |
| **Average interview length** | 63 min (varying between 37 to 90 minutes) |
| **No. of (vice) principals** | 6 |
| **No. of teachers (special education/resource/subject)** | 21 |
| **No. of guidance counselors** | 7 |
| **No. of school social workers** | 5 |
| **No. of school psychologists** | 3 |
| **No. of school nurses** | 6 |
| **No of school doctors** | 4 |
| **No. of social workers** | 3 |
| **No. of resource teachers** | 1 |

Various health-related reasons for absenteeism were recognized, the most common being a general somatic health issue. Many participants perceived medical absences difficult to intervene in and thus a risk for continued absenteeism. However, health reasons were also considered valid reasons for absenteeism and their detrimental effects could be ameliorated by support from home. The participants also mentioned several leisure-related reasons for absenteeism. The ideation of the student towards school could also promote absenteeism.

A number of issues related to the family were identified (Table 2). The situation of the parent/guardian could be the cause of absenteeism, as could life changes that the family is undergoing.

Absenteeism was thought to mirror how much the student enjoys school. The participants identified both specific, such as relationships within the school environment, and non-specific school-related reasons that could cause absenteeism.

**Table 2. Classification of the reasons for absenteeism according to the participants of this study.**

| **1. Student-related reasons** | | | | | |
|---|---|---|---|---|---|
| *Health-related reasons* | General somatic problems | Headache/stomach pain | General mental health problems | Depression/anxiety | Inability to leave from home | Learning difficulty |
| *Leisure* | Friends | Gaming | Hobbies | Substance use | | |
| *Ideation* | Truancy | Motivational issues | General attitude towards school | | | |
| **2. Family-related reasons** | | | | | |
| *Family problems* | General problems in the family | Problems in family's interaction | | | | |
| *Parent/guardian-related reasons* | Attitude towards school | Insufficient parenting skills | Day routine | Parent's/guardians unemployment/health issues | Ability to estimate health | |
| *Changes in the family* | Family crisis | Immigration | | | | |
| *Other aspects of family life* | Travel | Religion | | | | |
| **3. School-related reasons** | | | | | |
| *Relationships within the school environment* | Bullying | Relationship with teacher | Relationships within class | | | |
| *Unspecified aspects of school life* | Middle school culture | Electronic student management system | | | | |

## Reasons promoting the inclusion of SHC in absenteeism reduction

Educational professionals, SHC professionals, and social workers thought that SHC should be included in absenteeism reduction when the absences are primarily health-related. Educational professionals emphasized that they have no health care training and thus valued the opinion of SHC professionals in these situations. Educational professionals thought that it might be easier for the student to talk about physical symptoms rather than about mental health problems whereas social workers pondered whether it would be easier for parents/guardians to accept help offered by a health service rather than child protection services.

"*When there are anxiety symptoms, there should be an appointment with the school doctor, so the school nurse makes the appointment and the guardian is offered this option.*" *(principle, FG10)*

"*And in my opinion this sounds like a very good structure [where a student is routinely referred to SHC after 50 hours of absences], and I also think that health services are often easier for the parents to accept, too, than getting a phone call about being reported to child protective services.*" *(social worker, FG9)*

Both educational professionals and social workers expressed that if the reason for absences was known to be a mental health issue, the school doctor should be included to assess the need of a referral to specialized medical care. They thought that the school nurse was a good partner in these situations as the nurse is present at the school more often than the school doctor. Besides, the school nurse often participates in school welfare group meetings, which the educational professionals valued as it facilitated approaching the nurse.

"*And [there is] a lot of co-operation, so that if I as a school social worker have met with a student and I feel that they might be in need of adolescent psychiatry or some other referral, then once a week we have a school doctor present and an appointment can be booked through the school nurse.*" *(school social worker, FG10)*

"*I think that the teachers' lounge functions really well; there you can see the school nurse and there is an exchange of information and worries and thoughts.*" *(teacher, FG1)*

School doctors perceived intervening in absenteeism as part of their job and considered themselves good partners in these situations, especially since, due to the health checks they routinely perform, they have a comprehensive understanding of the situation of the student. The doctors reported that they are infrequently able to participate in the school welfare group meetings. Thus they perceived themselves less affiliated with the school and therefore possibly better able to build rapport with the family. If the school nurse was well-integrated with the school, the doctors felt more knowledgeable about the absences of a particular student. The authority associated with health care workers also promoted their role in absenteeism reduction according to both nurses and doctors.

"*So if you work as a full-time school doctor, this is everyday life. Absences are a part of the everyday. They are a part of the job. I think that it's one of the most important aspects of the job.*" *(school doctor, FG6)*

"*[Privacy and confidentiality] are felt to be very strong [in SHC], so that if the family has issues that they don't want the school to know about, it might be easier for them to talk about*

*these issues with the school doctor, specifically, whom they might perceive to be a bit on the outside compared to rest of the school . . ." (school doctor, FG6)*

*"Our place is towards the end of the line, so many others—or the school nurse has first looked into it and so on, so maybe you feel like the school doctor could kind of use their authority a little and tell the guardians at an earlier stage that, really, if it's a mild headache or a little pinch in the tummy or perhaps not even a pinch . . . then [the guardians] should just kind of nudge the child to school and maybe the child will start to feel better." (school doctor, FG3)*

### Reasons inhibiting the inclusion of SHC in absenteeism reduction

Educational professionals, SHC professionals, and social workers were all concerned about the inability of SHC to access the electronic school records and thus being unaware of any absences. There was also a shared concern for SHC resources. Specifically, both educational professionals and school nurses perceived time constraints regarding the schedule of the nurse, wishing for more time per student as well as time to participate in health education.

*"I feel that it is especially problematic, even worrying, that we have this problem—apparently originating in legislature—that health care workers are unable to access a student's [electronic student management system] record, because it could be an especially important factor in strengthening the offered support if health care workers could immediately, in real time, see things there." (teacher, FG1)*

*"Right now, at least, we have a doctor that is very interested in students who are often absent or otherwise have problems, but sometimes it's a question of resources; a school doesn't always have a school doctor, or then there might be more focus on broad health check-ups or on those who need a referral." (nurse, FG1)*

The educational professionals brought up a number of issues related to the main focus of SHC. They were unsure whether SHC would even be interested in absenteeism, and suspected that the interest might pertain to a certain doctor instead of being essential in their job description. Some were dissatisfied with the perceived lack of co-operation of SHC, primarily because doctors rarely participated in the school welfare group meetings. Perceived differences in work cultures further hindered collaboration. School doctors were considered a slow route to help, and were often unfamiliar among school personnel as they tend to change often. The educational professionals were sceptic of any possible effects of SHC on absenteeism.

*"I would say that the school doctor, maybe even the school nurse, do somehow remain pretty unknown to the majority of students. Naturally, the nurse is more familiar than the doctor, because certain students visit the nurse quite often. But maybe in a situation where one starts to examine [the student's] absenteeism or the problems behind it, we should perhaps include an adult who has a connection with the student." (special education teacher, FG7)*

*"SHC services are not just about health check-ups. Currently, school nurses do this kind of unauthorized basic work—or the kind of work that is not considered work output—when they participate in these student consultations or the meetings of the student welfare group." (school social worker, FG2)*

Social workers were likewise concerned whether absenteeism was within the focus of SHC. Furthermore, they doubted whether school doctors would participate in a meeting with them if invited.

*"So, this multi-professional needs assessment I talked about before the interview: In the future, we will include the referring party more in the assessment, so for instance from school that might be the school psychologist, school social worker, or a teacher, or whoever is considered most important; I suspect that doctors will probably be difficult to get to participate, but school nurses might participate more." (social worker, FG8)*

School nurses preferred to focus on health checks and worried about becoming burdened by any absenteeism reduction efforts. The majority of them felt that parents/guardians delegated their duties, like assessing the capability of a student to attend school, to the nurses, thus increasing the workload of the nurses.

*"As I listen to everyone here I get the feeling [that people are saying] that 'let's burden the nurse more', all the time, more, more, and more, so that you should be a psychiatric nurse and an acute care specialist and, in a way, the whole package, and at the same time manage the entire social side, to give advice on who to contact and when it's needed." (nurse, FG6)*

School doctors stated that they are unable to participate in school welfare group meetings since meetings are often scheduled without considering their schedule. Doctors asserted that their inclusion in absenteeism reduction should be independent of their participation in welfare group meetings. The doctors recognized some collaboration difficulties due to inadequate facilities which may force the nurse and doctor to work at a particular school on different days and to communicate either via telephone or e-mail. They agreed that doctors tend to change frequently and thus the educational professionals might be unfamiliar with the current doctor.

*"It wouldn't work for us to be invited to every meeting, but when discussing the subject during the meeting, our role should also be considered." (school doctor, FG5)*

Doctors also mentioned that a change in legislation had precluded the educational professionals from consulting the doctor as freely as they used to since currently only anonymous consultations are permitted without the permission of the respective student. Furthermore, doctors also claimed that school nurses had a distinctly different role from doctors and that one professional is unable to substitute the other.

*"It's this law that causes it. [. . .] There is uncertainty about whether I am allowed to ask about this or that, or can I talk about it, and . . . Then you ask, you consult anonymously—which is completely silly, when the subject is a child's medical absences—when I could just get on the computer and see whether there are health issues in the background." (school doctor, FG5)*

*"School nurses and school doctors don't do the same work, and they don't have the same perspective. Our educations are really totally different. The nurse isn't a "little doctor," and we can't act as surrogate nurses; we have separate jobs and different tasks, and the expertise of both is needed." (school doctor, FG5)*

## Discussion

This qualitative study reports on focus group interviews organized to better understand the reasons for absenteeism in middle school according to different professionals and the perceived role of SHC in absenteeism reduction. Participants encounter absentee students regularly due to their work (different educational professionals, SHC professionals, and social

workers from child protective services). A number of reasons for absenteeism were named, and absenteeism appeared to occur in every school, despite different socioeconomic areas. Although reported in previous studies (8–11), societal reasons were not mentioned.

The most important reason supporting the inclusion of SHC in absenteeism reduction is the fact that health issues were both the most commonly named reason for absenteeism and an issue which the educational professionals had difficulties addressing. Furthermore, the school doctors expressed their wish to be included in these processes. Additionally, scientific evidence supports the concept that SHC measures can reduce absenteeism [6, 26–28]. SHC is provided within the everyday environment of students which supports the effectiveness of SHC measures in absenteeism reduction [24]. For example, this proximity minimizes the transport needs of students and allows rapid reactions to the ever-changing circumstances of students. However, both educational professionals and social workers thought that the main focus of SHC was routine health checks, and they questioned the resources, interest, and general effectiveness of SHC in absenteeism reduction.

The study participants recognized the same reasons for absenteeism that have been reported in previous studies [1, 7, 11, 17, 47]. However, they stated more student- and family-related than school-related reasons for absenteeism. In the present study, a general somatic health complaint was the most commonly named reason for absenteeism. By contrast, when given a list of possible reasons for absenteeism, participants in a Swedish study chose an adverse home situation as the most common reason for absenteeism, and somatic complaints were only ranked 11[th] [17]. The Swedish research group was concerned whether teachers were familiar with the association between school absenteeism and mental health problems [17]. The findings of the present study suggest that Finnish educational professionals are well aware of this connection since mental health issues were a reason to refer the student to the school doctor. In previous studies, students and parents/guardians have been more adept than educational professionals in identifying school-related factors for absenteeism [48, 49].

No societal reasons for school absences were mentioned. One possible explanation is the equity in Finnish society and education, depicted by the Gini coefficient [50], and the uniform performance of Finnish schools in the PISA evaluations [34]. Some aspects of the Finnish educational system, such as free basic education, may ameliorate some of the socio-economic factors associated with absenteeism. Other neighborhood aspects, like gang activity and unsafety of certain neighborhoods [12, 13], may be less relevant in Finland than in other countries.

Since previous studies [6, 26–28] have supported the ability of SHC to reduce absenteeism, the reasons that inhibited the inclusion of SHC in absenteeism reduction deserve special consideration. Based on our findings, a lack of communication between different professionals seems apparent and may cause faulty assumptions. This is depicted by the school nurses who were hesitant to include SHC in absenteeism interventions. Regular contact between educational professionals and school doctors seems warranted as this would familiarize professionals with each other and create a natural opportunity for consultations. Based on our findings, policy makers should promote the collaboration of different professionals in Finnish schools. The efficacy of SHC could possibly be improved through systematically shared information regarding student absences, which could in turn expedite interventions.

School absenteeism is a global problem that threatens the development of youth worldwide. International studies have identified similarities in the structure of school health care in different countries [19]. Since the education and health sectors are often governed separately, the obstacles hindering the inclusion of SHC in absenteeism reduction are also plausibly similar internationally.

The focus group method is suitable for studying how the public experiences health care, allowing the discovery of both what the participants think of the subject, and why they think a

certain way [51–53]. Absenteeism reduction requires collaboration between different professionals which unofficial policies and subjective perceptions may hinder. Focus groups can uncover such obstacles, so they were chosen as the research method. Based on the pilot focus groups, we decided that the groups could be heterogenous in composition, including participants from different professions, in an attempt to enhance rich interaction. The groups were predominantly "naturally occurring" in composition, as recommended [51]. We respected the principles of qualitative research, such as Lincoln and Guba's Evaluative Criteria [54] and the COREQ checklist [46]. We were able to avoid moderator bias, a stereotypical limitation of focus groups interviews [55], by choosing an experienced moderator with no foreknowledge on SHC. Credibility was ensured with both triangulations and repeated discussions of three researchers about the data and its categorization. We used thick description of the research process, study sites, and participant selection to ensure transferability. Dependability was established with the pilot-tested interview guide. Data saturation was repeatedly discussed during data collection. Confirmability was established with both reflexivity and describing the analysis process minutely.

One school declined to participate in this study based on the claim that absenteeism was not an issue in their school. Several research projects were conducted in Helsinki concurrently with this study. Thus, schools had to choose which studies to participate in. Other studies may have burdened the schools that chose not participate.

This study has some limitations. Firstly, school doctors and social workers were interviewed in a location outside their workplace during working hours so they had to organize their timetable to enable participation. This might have reduced the number of possible participants. Secondly, our focus groups were conducted in southern Finland with the highest population density in the country; interviews in a rural environment might have provided different answers. Thirdly, three members of the research group had previously worked in SHC so they may have had preconceptions of the subject. To avoid bias, we recruited an interviewer who had used this method previously and had no experience in SHC. With regard to applicability, the usual issues of qualitative research exist. The exploratory nature of data collection may affect the generalizability of the findings. Despite the seemingly low number of participants, saturation was reached which implies that the number of participants was sufficient. Educational and health systems vary internationally, and this study was conducted prior to the Covid-19 pandemic. Although the focus groups were organized between 2016–2017, national policies regarding school absenteeism remain unchanged. The Covid-19 pandemic has globally exacerbated adolescents' mental health problems [56], which highlights the relevance of all possible measures to alleviate them.

## Conclusion

Different professionals working in schools recognize the varied origins of school absenteeism. Including SHC in absenteeism interventions, however, elicits differing perceptions. Global efforts are warranted to ascertain the role of SHC in absenteeism interventions. Local authorities should strive to smoothen the collaboration of professionals so that absentee students receive the support they need to counter the myriad of reasons resulting in absenteeism.

## Supporting information

**S1 Appendix. Focus group discussion guide.**
(DOCX)

**S1 Data. Reasons for absenteeism.**
(DOCX)

## Acknowledgments

We want to thank each professional who participated in this study.

## Author Contributions

**Conceptualization:** Katja Melander, Tiina Kortteisto, Elina Hermanson, Riittakerttu Kaltiala, Katariina Mäki-Kokkila, Minna Kaila, Silja Kosola.

**Data curation:** Katja Melander, Tiina Kortteisto.

**Formal analysis:** Katja Melander, Tiina Kortteisto.

**Funding acquisition:** Katja Melander.

**Investigation:** Katja Melander.

**Methodology:** Tiina Kortteisto.

**Project administration:** Minna Kaila, Silja Kosola.

**Resources:** Katja Melander.

**Software:** Katja Melander.

**Supervision:** Minna Kaila, Silja Kosola.

**Visualization:** Katja Melander.

**Writing – original draft:** Katja Melander.

**Writing – review & editing:** Katja Melander, Tiina Kortteisto, Elina Hermanson, Riittakerttu Kaltiala, Katariina Mäki-Kokkila, Minna Kaila, Silja Kosola.

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
