## [Decision Letter · Decision Letter 0]

28 Sep 2021

PONE-D-21-17857The perceptions of different professionals on school absenteeism and the role of school health care. A focus group study.PLOS ONE

Dear Dr. Melander,

Thank you for submitting your manuscript to PLOS ONE. After careful consideration, we feel that it has merit but does not fully meet PLOS ONE’s publication criteria as it currently stands. Therefore, we invite you to submit a revised version of the manuscript that addresses the points raised during the review process.

We look forward to receiving your revised manuscript.

Kind regards,

Webster Mavhu

Academic Editor

PLOS ONE

Journal Requirements:

3. Please include your tables as part of your main manuscript and remove the individual files. Please note that supplementary tables (should remain/ be uploaded) as separate "supporting information" files.

Additional Editor Comments (if provided):

-Please accept our apologies for the delay in reaching a decision on your manuscript. In addition to reviewers comments, address the following:

-It is unclear what the objective of this study was from a Public/Global Health perspective.

-I am concerned that this study was conducted about 4 years ago. Are the findings still applicable? e.g. When the focus groups were conducted, basic education ended after nine years of basic education or at age 17 (line 86-87). Is this still the same or has changed? If this has changed, what are the implications of the findings.

-The focus groups were organized between spring 2016 and winter 2017 and focus groups were digitally recorded and transcribed verbatim in 2018. What did "organizing" actually entail considering it took nearly a year? What is the reason behind the time lag between "organizing" and conducting (winter of 2017 and 2018). Perhaps it would help to simply say when the FGDs were conducted instead of saying when they were organized e.g. focus groups were digitally recorded from January to February 2018.

-Abstract - Data analysis was both inductive and deductive (line 31-32). All identified themes were derived from the recorded data (line 189). The latter assumes an entire inductive approach?

-Total number of participants was 55. On average, each FGD had about 6 participants? A FGD generally comprises 8-12 participants. This ensures validity of findings. In results, include a range of FGD participants e.g. 6-10. Also, comment on relatively low number of participants under limitations. Also, the FGD's inherent limitations should be mentioned.

-Still, it is unlikely that a study with just 55 participants could influence policy even if at national level? Discussion could end with a recommendation of additional/larger studies?

-Move lines 71-76 to Methods (lines 180-186). Introduction could end instead with why this study was conducted e.g. to inform XX interventions.

-Leave out lines 149-151.

Reviewers' comments:

Reviewer's Responses to Questions

**Comments to the Author**

1. Is the manuscript technically sound, and do the data support the conclusions?

Reviewer #1: Yes

Reviewer #2: Partly

2. Has the statistical analysis been performed appropriately and rigorously? 

Reviewer #1: N/A

Reviewer #2: N/A

3. Have the authors made all data underlying the findings in their manuscript fully available?

Reviewer #1: Yes

Reviewer #2: Yes

4. Is the manuscript presented in an intelligible fashion and written in standard English?

Reviewer #1: Yes

Reviewer #2: Yes

5. Review Comments to the Author

Reviewer #1: My comments also in a separate file.

The perceptions of different professionals on school absenteeism and the role of school health care. A focus group study.

Number PONE-D-21-17857

This is an interesting and well designed qualitative research. Its aim is to assess the role of school health services in decreasing absenteeism among adolescent pupils 13 to 15 years in Finland. The potential limit of the paper is that the results may not apply to other countries, but this problem is counter-balanced by the fact that the authors provide a thorough description of the context of education, schooling and school health in their country. The paper should be revised and specify how the issue of absenteeism has been operationalized: it is different to skip school once or twice in a year for a flu or a broken leg, or to repeatedly miss periods and days for vague reasons or for reasons that are even not provided. Also, the authors rightly mention the issue of “drop-outs”, that is youngsters who do not appear anymore for long periods or definitively, but do not tackle this issue specifically.

The introduction and the methods sections should thus be modified accordingly. It would be as well useful to know what the content is of the yearly nurses’ check-ups; same applies to the doctors’ check-ups.

The analysis of the data is carefully described and meets the standards of a qualitative research.

I have no comments on the results.

The discussion is well presented. It would be useful to comment on the lessons learned for readers from outside Finland regarding the role of school health services. Also, I miss a few remarks on what the authors plan to do, as they point out that there is room for improvement in this area. I think, but this is a personal opinion, that many school health services spend a lot of time on screening procedures whose effectiveness is not necessarily evidence-based. They may thus devote more time to the support given to vulnerable pupils and health promotion. But the authors don’t need to comment on this issue.

Reviewer #2: Thank you for the opportunity to review the article, “The perceptions of different professionals on school absenteeism and the role of school health care - A focus group study.” This study addresses important topics, specifically the perceptions of Finnish education professionals of the reasons students miss school and the potential role - if any - of Finnish school health care (SHC) in mitigating student absenteeism. While there are some merits to the article with respect to originality, research questions, and findings, the quality of the research design does not meet the high standards of a journal like PLoS ONE. I hope that my comments and suggestions are helpful to the authors as they revisit the manuscript. If the authors intend to resubmit this for review, I recommend significant revisions to both the introduction and methods. I detail my concerns below.

Introduction

The introduction seems incomplete. I feel the authors haven’t adequately positioned their research in the absenteeism literature or policy context - i.e., why should I care about this study? What is the Finnish policy context? Why is absenteeism of such concern? What gaps does this study fill in the extant literature? More specific to this study, why might we have reason to believe that SHC has an important role to play in mitigating absenteeism?

Research Questions

Research questions 1 and 3 are interesting, but I think they would be more compelling if the aforementioned introduction adequately made a better case for research question 2, which the authors hint at in the final paragraph of the introduction.

Method

The study focus is somewhat narrow, looking only at 13-15 year olds. The authors provide some evidence that is when absenteeism usually increases, but the research questions address the issue more broadly. The reasons 13-15 year olds miss school are likely not the same reasons older or younger students miss. That is not to say that the narrow study design is not worthwhile, merely that it does not answer the research questions as currently constructed. Further, there is no mention of this age group in the literature review.

The rationale for the selection of a focus group design is not made clear by the authors. Focus groups can obviously be a useful method, but rarely are used as a stand-alone technique. I am concerned about the credibility of participant responses when answering within a focus group setting. Ideally, the researchers would be able to follow up with at least some of the participants one-on-one to ensure their responses were adequately forthcoming and truthful. In lines 429-431, the authors write that “Based on the pilot focus groups, we decided that the groups could be heterogenous (sic) in composition, including participants from different professions, in an attempt to enhance rich interaction.” While participant interaction is certainly a valuable feature of focus groups, it does not adequately address my concerns about participants’ willingness to give the most honest and complete responses. I certainly appreciate the authors’ attention to credibility through triangulation and discussion, but those strategies address the credibility of the data already collected, rather than ensure that the highest quality data are collected in the first place.

Results

The findings of this study are interesting and carry clear policy implications for the role of SHC in improving student attendance, especially regarding improving communication between school staff and school health professionals. If the issues above are addressed, I think this paper could be a valuable contribution to the absenteeism literature. The authors may want to consider how their findings fit in into the existing literature, for example “Absent from School: Understanding and Addressing Student Absenteeism” edited by Gottfried and Hutt (which has a chapter on school-based health centers).

6. PLOS authors have the option to publish the peer review history of their article (what does this mean?). If published, this will include your full peer review and any attached files.

Reviewer #1: **Yes: **Prof. Pierre-Andre Michaud, Lausanne University

Reviewer #2: No

---

## [Author Response · Author response to Decision Letter 0]

17 Nov 2021

Journal requirements and editor comments

We have followed the PLOS ONE guidelines, including style and file naming.

In your Data Availability statement, you have not specified where the minimal data set underlying the results described in your manuscript can be found. PLOS defines a study's minimal data set as the underlying data used to reach the conclusions drawn in the manuscript and any additional data required to replicate the reported study findings in their entirety. 

Thank you for expressing your concern. We have included all the relevant data within the manuscript. Because the focus groups were conducted in Finnish, full transliterations are available from the first author upon request.

Please include your tables as part of your main manuscript and remove the individual files. Please note that supplementary tables (should remain/ be uploaded) as separate "supporting information" files.

We have included our tables as part of the manuscript as suggested. 

Please include captions for your Supporting Information files at the end of your manuscript, and update any in-text citations to match accordingly. 

We have added captions as requested. 

It is unclear what the objective of this study was from a Public/Global Health perspective.

Thank you for your comment. School absenteeism is a global problem that jeopardizes the health and development of the student since it associates with risky behavior. Extensive absenteeism also threatens the school performance of the student thus lowering the educational level achieved. Both of these reasons may lead to inferior health in adulthood. We have added information on the possible repercussions of excessive absenteeism (please see lines 47-53). 

More than 100 countries have school health services and intervening in school absenteeism is one of the goals of school health care work by international standards (see lines 75-77). However, the organizational models of school health care differ between countries. Often collaboration between professions may be hindered by unofficial policies and subjective perceptions. Focus groups may be used to recognize these hidden obstacles and the results may help in overcoming them. We have clarified the aims of the study in the manuscript (please see lines 69-70 and 87-90).

I am concerned that this study was conducted about 4 years ago. Are the findings still applicable? e.g. When the focus groups were conducted, basic education ended after nine years of basic education or at age 17 (line 86-87). Is this still the same or has changed? If this has changed, what are the implications of the findings.

Thank you for expressing your concern. The study findings are still relevant because students still need to apply to high school or vocational education (i.e. years 10-12), and this secondary education is most often provided separately from elementary/primary education. Years 7-9 (ages 13-15 years, respectively) are still the time when school absences peak. Since autumn 2021, compulsory education lasts for 12 years as stated in the manuscript. The aim of this change was, among other goals, to ameliorate the impact absenteeism has on the school performance of an absentee student. This is the only major structural change in our educational system that has occurred since the data was gathered. No other major changes have been undertaken on a national level regarding intervening in absenteeism. Additionally, gathering evidence implies that the Covid-19 pandemic has exacerbated absenteeism, making this report all the more relevant. We have amended the study limitations section (see lines 508-512) and discussed the time lapsed from the gathering of the data. 

The focus groups were organized between spring 2016 and winter 2017 and focus groups were digitally recorded and transcribed verbatim in 2018. What did "organizing" actually entail considering it took nearly a year? What is the reason behind the time lag between "organizing" and conducting (winter of 2017 and 2018). Perhaps it would help to simply say when the FGDs were conducted instead of saying when they were organized e.g. focus groups were digitally recorded from January to February 2018.

Thank you for your helpful suggestion. The organization process entailed acquiring a research permit, contacting the schools/professionals of interest, having an unofficial preliminary meeting with the representative of the school/professionals, and finally organizing the focus groups. We have elaborated on the organization process (see lines 180-195).

Abstract - Data analysis was both inductive and deductive (line 31-32). All identified themes were derived from the recorded data (line 189). The latter assumes an entire inductive approach?

We appreciate your observations. We have clarified our data analysis (please see lines 32-33). The analysis was mainly inductive but the categorization of our findings was based on existing literature, giving the analysis deductive features as well.

Total number of participants was 55. On average, each FGD had about 6 participants? A FGD generally comprises 8-12 participants. This ensures validity of findings. In results, include a range of FGD participants e.g. 6-10. Also, comment on relatively low number of participants under limitations. Also, the FGD's inherent limitations should be mentioned.

Thank you for your comments. The range of FGD participants is mentioned on line 237. We feel that a limited number of participants guaranteed a safe environment where the participants could speak freely. In addition, the smaller sizes of the FDGs ensured that every opinion could be heard. Some sources also recommend limiting the number of participants to 10 people to ensure that each participant has the chance to share their thoughts. As saturation was reached, we deemed organizing additional focus groups unnecessary even though our research permit would have allowed it. Based on the comments of the editor and the referees, we have developed the discussion on the strengths and limitations of our study (see lines 476-491 and 498-512); for instance, we discuss the number of participants (see lines 507-508) as well as the inherent limitations of the focus group method (see lines 485-486 and 506-507).

Move lines 71-76 to Methods (lines 180-186). Introduction could end instead with why this study was conducted e.g. to inform XX interventions. Leave out lines 149-151.

We have made the suggested changes. 

Comments of the first reviewer

This is an interesting and well designed qualitative research. Its aim is to assess the role of school health services in decreasing absenteeism among adolescent pupils 13 to 15 years in Finland. The potential limit of the paper is that the results may not apply to other countries, but this problem is counter-balanced by the fact that the authors provide a thorough description of the context of education, schooling and school health in their country. The paper should be revised and specify how the issue of absenteeism has been operationalized: it is different to skip school once or twice in a year for a flu or a broken leg, or to repeatedly miss periods and days for vague reasons or for reasons that are even not provided. Also, the authors rightly mention the issue of “drop-outs”, that is youngsters who do not appear anymore for long periods or definitively, but do not tackle this issue specifically. The introduction and the methods sections should thus be modified accordingly.

Thank you for your insightful comments. To ensure that the article is useful to an international audience, we have added recommendations on how school health care should be organized according to international standards and compared the Finnish school health care system to the composition of school health care in other countries (see lines 75-83). We have also clarified our definition of school absenteeism (please see lines 70-73). Since dropouts are rare in Finland, they were not the focus of this article. We have clarified this in the revised manuscript (see lines 117-120).

It would be as well useful to know what the content is of the yearly nurses’ check-ups; same applies to the doctors’ check-ups.

We have added information on the content of the check-ups (see lines 135-139).

The analysis of the data is carefully described and meets the standards of a qualitative research.

Thank you for this reassurance.

The discussion is well presented. It would be useful to comment on the lessons learned for readers from outside Finland regarding the role of school health services. Also, I miss a few remarks on what the authors plan to do, as they point out that there is room for improvement in this area. I think, but this is a personal opinion, that many school health services spend a lot of time on screening procedures whose effectiveness is not necessarily evidence-based. They may thus devote more time to the support given to vulnerable pupils and health promotion. But the authors don’t need to comment on this issue.

Thank you for your perceptive remarks. We thoroughly agree that school health care should focus on evidence-based methods instead of all-encompassing but inefficient screening. In our discussion we have added a paragraph on the meaning of our findings from an international perspective (see lines 470-474). Additionally, we have made suggestions on how the system in Finland should be developed based on our findings (see lines 459-468).

Comments of the second reviewer

Thank you for the opportunity to review the article, “The perceptions of different professionals on school absenteeism and the role of school health care - A focus group study.” This study addresses important topics, specifically the perceptions of Finnish education professionals of the reasons students miss school and the potential role - if any - of Finnish school health care (SHC) in mitigating student absenteeism. While there are some merits to the article with respect to originality, research questions, and findings, the quality of the research design does not meet the high standards of a journal like PLoS ONE. I hope that my comments and suggestions are helpful to the authors as they revisit the manuscript. If the authors intend to resubmit this for review, I recommend significant revisions to both the introduction and methods. I detail my concerns below.

Thank you for your encouraging words. We have tried to answer your concerns to the best of our ability. 

The introduction seems incomplete. I feel the authors haven’t adequately positioned their research in the absenteeism literature or policy context - i.e., why should I care about this study? What is the Finnish policy context? Why is absenteeism of such concern? What gaps does this study fill in the extant literature? More specific to this study, why might we have reason to believe that SHC has an important role to play in mitigating absenteeism?

Thank you for expressing your concerns. We have modified the introduction to clarify the possible repercussion of school absenteeism to better justify why absenteeism needs to be taken seriously (see lines 47-53). We have also explained why health care professionals should be included in absenteeism reduction (see lines 75-88). Furthermore, we have touched upon the Finnish policy context in the introduction (see lines 81-83). To our knowledge, no previous study has explored factors either promoting or inhibiting the inclusion of school health care in absenteeism reduction (the aim stated on lines 87-88). Based on existing literature the ability of school personnel to recognize the reasons behind absenteeism seem to vary; we aim to clarify this controversy (see lines 63-70). 

When it comes to the logic promoting the inclusion of school health care in absenteeism interventions, previous studies with diverse settings have indicated that school health care measures can in fact be useful in absenteeism reduction (please see lines 86-88). Also, school health care is provided in the everyday environment of the students, which increases its potential effectivity on students in need (as mentioned on lines 431-434). In addition, absenteeism is often a consequence of either somatic symptoms and/or mental health issues (see lines 77-80) which school health care staff are better equipped to assess and treat than educational staff.

Research questions 1 and 3 are interesting, but I think they would be more compelling if the aforementioned introduction adequately made a better case for research question 2, which the authors hint at in the final paragraph of the introduction. 

Thank you for your insightful comments. We have reformulated our research questions to better match the study setting (see lines 96-99) in addition to revisions of the introduction.

The study focus is somewhat narrow, looking only at 13-15 year olds. The authors provide some evidence that is when absenteeism usually increases, but the research questions address the issue more broadly. The reasons 13-15 year olds miss school are likely not the same reasons older or younger students miss. That is not to say that the narrow study design is not worthwhile, merely that it does not answer the research questions as currently constructed. Further, there is no mention of this age group in the literature review.

We reviewed the gathered literature again and highlighted some studies that discuss the predictive value of school absenteeism in this age group (see lines 88-90).

The rationale for the selection of a focus group design is not made clear by the authors. Focus groups can obviously be a useful method, but rarely are used as a stand-alone technique. I am concerned about the credibility of participant responses when answering within a focus group setting. Ideally, the researchers would be able to follow up with at least some of the participants one-on-one to ensure their responses were adequately forthcoming and truthful. In lines 429-431, the authors write that “Based on the pilot focus groups, we decided that the groups could be heterogenous (sic) in composition, including participants from different professions, in an attempt to enhance rich interaction.” While participant interaction is certainly a valuable feature of focus groups, it does not adequately address my concerns about participants’ willingness to give the most honest and complete responses. I certainly appreciate the authors’ attention to credibility through triangulation and discussion, but those strategies address the credibility of the data already collected, rather than ensure that the highest quality data are collected in the first place.

Thank you for expressing you concerns. Often collaboration between professions may be hindered by unofficial policies and subjective perceptions. Focus groups may be used to recognize these hidden obstacles and the results may help in overcoming them. During the focus group sessions, the mood was relaxed and the participants were forthcoming with their views. The groups were “naturally occurring” by composition, a feature recommended by previous authors. 

The groups were small in size, guaranteeing that the thoughts of each participant were taken into account. All participants engaged in this study voluntarily. Most of the participants felt that there was room for improvement in absenteeism reduction processes which motivated their participation in the current study. Additionally, PlosOne has previously published articles with focus groups with fewer participants than ours (for instance https://doi.org/10.1371/journal.pone.0191635 and https://doi.org/10.1371/journal.pone.0228054). 

The findings of this study are interesting and carry clear policy implications for the role of SHC in improving student attendance, especially regarding improving communication between school staff and school health professionals. If the issues above are addressed, I think this paper could be a valuable contribution to the absenteeism literature. The authors may want to consider how their findings fit in into the existing literature, for example “Absent from School: Understanding and Addressing Student Absenteeism” edited by Gottfried and Hutt (which has a chapter on school-based health centers).

Thank you for your helpful comments and encouraging words. We have familiarized with the suggested book and added it as a reference (please see lines 79-80, 88-90, and 431-434).

---

## [Editor Report · Decision Letter 1]

26 Dec 2021

PONE-D-21-17857R1The perceptions of different professionals on school absenteeism and the role of school health care. A focus group study.PLOS ONE

Dear Dr. Melander,

Thank you for submitting your manuscript to PLOS ONE. We are about to accept the manuscript. Please address the following:

Consider changing title to: The perceptions of different professionals on school absenteeism and the role of school health care: A focus group study conducted in Finland 

Full form of SHC should be given the first time abbreviation is used.

Would say "topic guide" instead of questionnaire.

We look forward to receiving your revised manuscript.

Kind regards,

Webster Mavhu

Academic Editor

PLOS ONE

Journal Requirements:

Additional Editor Comments:

Consider changing title to: The perceptions of different professionals on school absenteeism and the role of school health care: A focus group study conducted in Finland

Full form of SHC should be given the first time abbreviation is used.

Would say "topic guide" instead of questionnaire.

See a few edits/suggestions in attached.
---

## [Author Response · Author response to Decision Letter 1]

29 Jan 2022

We thank the reviewers for their effort in improving our manuscript. We have heeded the suggestions made by the editor and ensured that the references need no revisions.

---

## [Editor Report · Decision Letter 2]

8 Feb 2022

The perceptions of different professionals on school absenteeism and the role of school health care. A focus group study conducted in Finland.

PONE-D-21-17857R2

Dear Dr. Melander,

We’re pleased to inform you that your manuscript has been judged scientifically suitable for publication and will be formally accepted for publication once it meets all outstanding technical requirements.

Kind regards,

Webster Mavhu

Academic Editor

PLOS ONE
---

## [Editor Report · Acceptance letter]

14 Feb 2022

PONE-D-21-17857R2 

The perceptions of different professionals on school absenteeism and the role of school health care: A focus group study conducted in Finland. 

Dear Dr. Melander:

I'm pleased to inform you that your manuscript has been deemed suitable for publication in PLOS ONE. Congratulations! Your manuscript is now with our production department. 

Kind regards, 

on behalf of

Dr. Webster Mavhu 

Academic Editor

PLOS ONE